# Role of Advanced Glycation End Products as New Biomarkers in Systemic Lupus Erythematosus

**DOI:** 10.3390/ijms25053022

**Published:** 2024-03-05

**Authors:** Irene Carrión-Barberà, Laura Triginer, Laura Tío, Carolina Pérez-García, Anna Ribes, Victoria Abad, Ana Pros, Marcelino Bermúdez-López, Eva Castro-Boqué, Albert Lecube, José Manuel Valdivielso, Jordi Monfort, Tarek Carlos Salman-Monte

**Affiliations:** 1Rheumatology Department, Hospital del Mar, 08003 Barcelona, Spain; 2Medicine Department, Medicine Faculty, Universitat Autònoma de Barcelona, 08193 Bellaterra, Spain; 3Inflammation and Cartilage Cellular Research Group, Hospital del Mar Research Institute (IMIM), C/Dr. Aigüader 88, 08003 Barcelona, Spain; 4Clinical Expertise Unit (UEC) in Systemic Autoimmune Diseases and Vasculitis, Hospital del Mar, 08003 Barcelona, Spain; 5Grupo de Investigación Translacional Vascular y Renal, IRBLleida, 25198 Lleida, Spain; 6Departament de Medicina Experimental, Universitat de Lleida, 25198 Lleida, Spain; 7Departament d’Endocrinologia i Nutrició, Hospital Universitari Arnau de Vilanova, 25198 Lleida, Spain; 8Grup de Recerca Obesitat i Metabolisme (ODIM), IRBLleida, Universitat de Lleida, 25198 Lleida, Spain; 9Centro de Investigación Biomédica en Red de Diabetes y Enfermedades Metabólicas Asociadas (CIBERDEM), Instituto de Salud Carlos III (ISCIII), 28029 Madrid, Spain

**Keywords:** systemic lupus erythematosus, advanced glycation end products, cardiovascular disease, biomarkers

## Abstract

Advanced glycation end-products (AGEs) may play a relevant role as inducers in the chronic inflammatory pathway present in immune-mediated diseases, such as systemic lupus erythematosus (SLE). AGEs concentrations have been associated, with discrepant results to date, with some parameters such as disease activity or accrual damage, suggesting their potential usefulness as biomarkers of the disease. Our objectives are to confirm differences in AGEs levels measured by cutaneous autofluorescence between SLE patients and healthy controls (HC) and to study their correlation with various disease parameters. Cross-sectional study, where AGEs levels were measured by skin autofluorescence, and SLE patients’ data were compared with those of sex- and age-matched HC in a 1:3 proportion through a multiple linear regression model. Associations of AGEs levels with demographic and clinical data were analyzed through ANOVA tests. Both analyses were adjusted for confounders. AGEs levels in SLE patients were significantly higher than in HC (*p* < 0.001). We found statistically significant positive associations with SLE disease activity index (SLEDAI) and damage index (SDI), physician and patient global assessment, C-reactive protein, leukocyturia, complement C4, IL-6 and oral ulcers. We also found a negative statistically significant association with current positivity of anti-nuclear and anti-Ro60 antibodies. AGEs seem to have a contribution in LES pathophysiology, being associated with activity and damage and having a role as a new management and prognosis biomarker in this disease. The association with specific antibodies and disease manifestations may indicate a specific clinical phenotype related to higher or lower AGEs levels.

## 1. Background

Advanced glycation end-products (AGEs) are a set of compounds whose formation is a complicated molecular process resulting from the non-enzymatic interaction of reducing sugars and associated metabolites with peptides, proteins, and amino acids [1]. AGEs can accumulate under hyperglycaemic and pro-oxidative conditions, and it has been postulated that they have a role in inflammation.

The mechanisms of toxicity of AGEs are mainly related to two facts. On the one hand, glycation favors cross-links between the modified proteins, causing structural alterations and resulting in gradual deterioration in cell and tissue function and the generation of new immunological epitopes [2]. On the other hand, AGEs are recognized by their own receptor (RAGE), which is expressed in multiple cells from the immune system [3]. RAGE is divided into extracellular, transmembrane, and intracellular segments [4]. The interaction of AGEs with RAGE can activate the downstream nuclear factor kappa-B (NF-κB) signaling pathway and promote the secretion of several cytokines.

Soluble RAGE (sRAGE is variant of RAGE, a positively charged 48-kDa cleavage product from RAGE that keeps the ligand binding site but loses the other two domains [5]. sRAGE binding to ligands terminates intracellular signal transduction due to the loss of the transmembrane and intracellular fragments and inhibits the proinflammatory processes mediated by RAGE and its ligands by acting as a decoy which competitively binds to RAGE ligands [6]. sRAGE and not RAGE levels have been studied and linked to inflammation [7] as sRAGE is soluble and easy measurable, while RAGE is a cell–bound receptor and hence tissues are required for its measurement.

So far, more than 20 AGEs have been described in tissues [8]. Due to their stability, the most measured AGEs are serum or plasmatic Nε-(carboxymethyl)lysine (CML) and pentosidine. However, a part of the AGEs has the characteristic of being fluorescent, so it is possible to quantify them in a single measurement using an autofluorescence reader. This technique that measures accumulated AGEs in the skin, makes this assessment more appropriate to quantify the concentration of AGEs in an individual throughout their life than that of a single specific moment in relation to an acute process. So that, skin AGEs may better correlate with disease control, duration, and complications than serum AGEs [9]. As a validation method, it has been described that this autofluorescent measurement correlates with the concentration of AGEs, both fluorescent and not fluorescent, measured in skin biopsies [10]. Some of the advantages of measuring skin AGEs vs serum or plasmatic ones consist of having non-invasive, real-time data, easily available and affordable.

In systemic autoimmune diseases, such as systemic lupus erythematosus (SLE), increased AGEs formation can be expected, as inflammation is one of the hallmarks of the disease. Chronic inflammation in SLE appears to be associated with an intensified glycation process and the formation of AGEs, having higher values compared to healthy controls (HC) been demonstrated in some studies [11,12,13,14,15]. At the same time, AGEs are also involved in the generation of more inflammation and reactive oxygen species, creating positive feedback that enhances inflammation and AGEs levels.

Regarding atherosclerosis, AGEs have been linked to increased vascular rigidity and atherosclerosis [16,17,18]. In SLE, the presence of accelerated atherosclerosis that cannot be fully explained by traditional risk factors for cardiovascular disease is a well-recorded phenomenon [19]. Some studies have suggested that increased levels of AGEs might contribute to the development of this accelerated atherosclerosis in SLE and, therefore, could be used as early markers for cardiovascular disease in this pathology [14,15].

Lately, there has been increased attention on the potential of RAGE and AGEs to target chronic inflammatory diseases such as SLE. Some studies have expounded on their usefulness as biomarkers of SLE diagnosis and prognosis, their relationship with accelerated atherosclerosis, as well as their potential place as targets for new treatments. However, we find some controversial results in the literature, showing that more and better studies are needed to fully elucidate their role in SLE.

Taking into account that the relation between skin AGEs and SLE has only been reported in one previous paper, the purpose of this work is to try to elucidate the role of AGEs in SLE as potential biomarkers of the disease, as well as their application in routine clinical practice as a tool for improving the diagnosis, monitoring, and/or prognosis of the disease, or as surrogate markers for the assessment of cardiovascular risk in this population. Our study involved describing AGEs concentrations in SLE and comparing them to age- and sex-matched HC; searching for correlations between AGEs concentrations and SLE characteristics such as specific manifestations, indexes of activity or accrual damage, or patient reported outcomes (PROs); and finally, exploring AGEs relationship with cardiovascular disease and cardiovascular risk factors (CVRF).

## 2. Results

### 2.1. Characteristics of Patients and Controls

The differences between the 189 HC and 62 cases are shown in Table 1: HC had a higher BMI and a higher incidence of dyslipidemia (both in total cholesterol and low-density lipoprotein values), obesity, hypertension, and active smoking. Patients with SLE had higher AGEs values and creatinine concentrations.

### 2.2. Comparison of AGEs in SLE Patients vs. Healthy Controls

According to all of the data explored, the multivariate model was adjusted with age, smoking, dyslipidemia, creatinine. The model reported a statistically significant difference between SLE and HC in AGEs values, showing that AGEs values in SLE patients were 0.721 (95% confidence interval (CI) [0.566; 0.876]) units higher (*p* < 0.001) than HC. See Table 2 for the analysis of covariance of fixed effects and Appendix A for the effects graphic.

### 2.3. Characteristics of SLE Patients According to AGEs Levels: Bivariate Analysis

A total of 122 SLE patients were included. All of the variables that showed statistically significant differences according to AGEs tertiles in the bivariate analysis are depicted in Table 3, adjusted by age (*p*-value M1) and by both age and smoking (*p*-value M2). The demographic characteristics and other SLE variables of interest are detailed in Appendix A.

### 2.4. Correlations between AGEs and SLE Characteristics: Multivariate Analysis

After adjustment for confounding variables, several SLE characteristics showed associations with AGEs levels. First of all, two of the most important SLE disease indexes, SLE disease activity index (SLEDAI) and SLE damage index (SDI), were significantly associated with AGEs levels. While for the SLEDAI we found a progressive increase in AGEs values as the SLEDAI activity escalated (AGEs values in patients with moderate and severe activity were 0.2 (95% CI [0.0006; 0.4], *p* = 0.0493) and 0.52 (95% CI [0.177; 0.86], *p* = 0.003) units higher than patients in remission/mild, respectively, we only found differences in SDI between those with low (0–2) and high scores (5, 6) (AGEs values 0.717 (95% CI [0.139; 1.295], *p* = 0.0156) units higher). This association with disease activity is also reflected in both the physician global assessment (PGA) and the patient global assessment (PtGA). In those cases, values higher than 1 (PGA) or 3 (PtGA) were associated with an AGEs increase. PGA score of 1–2 and a PGA score higher than 2 had AGEs levels 0.033 (95% CI [0.058; 0.61], *p* = 0.018) and 0.39 (95% CI [0.094; 0.694], *p* = 0.01) units higher than patients with a PGA of 0, respectively; and patients with a PtGA score >3 had AGEs levels 0.26 (95% CI [0.063; 0.46], *p* = 0.01) units higher than patients with PtGA score ≤3.

Regarding serum biomarkers, we observed an increment in AGEs levels as C-reactive protein (CRP) and IL-6 increased, but significant differences were only detected between the 3rd and 1st tertile: 0.259 (95% CI [0.035; 0.48], *p* = 0.02) units higher for CRP and 0.352 (95% CI [0.1; 0.6], *p* = 0.006) for IL-6. The same tendency was observed in the level of leukocyturia (0.369, 95% CI [0.112; 0.626], *p* = 0.005) and C4 complement, although in this last one, significant differences with the 2nd tertile were also observed (0.25 (95% CI [0.02; 0.48], *p* = 0.0335) units higher for the 2nd tertile; and 0.28 (95% CI [0.056; 0.514], *p* = 0.015) for the 3rd one).

With reference to autoantibodies, a negative association was found between AGEs levels and both the presence of ANA or anti-Ro60 antibodies in the blood test performed for the study, where AGEs values were 0.496 (95% CI [0.937; 0.054], *p* = 0.028) and 0.26 (95% CI [0.5; 0.017], *p* = 0.035) units lower, respectively.

Finally, patients which had ever presented oral ulcers, a prevalent SLE manifestation, had AGEs values 0.216 (95% CI [0.02; 0.41], *p* = 0.03) units higher than patients who had never. All of these data are depicted, according to the prediction of each model, in Figure 1 and Figure 2 which graphically represent the mean and its corresponding 95% CI of AGEs for each category of variables. *p*-values < 0.05 indicate significant differences between the categories and the reference level of each variable. Also, the fixed-effects ANCOVA model between AGEs and each of the variables are provided Appendix A.

## 3. Discussion

We observed statistically significant differences between AGEs values measured by skin autofluorescence in SLE patients vs. HC. This difference has only been studied in two previous works [14,15] with small sample sizes (55 and 30 cases respectively, matched 1:1 with HC), and our research builds upon these studies in the following ways. First, we have increased the sample size, especially the HC sample, by matching cases with HC in a 1:3 proportion instead of a 1:1 proportion, making the study more robust. Secondly, we selected HC that had at least one CVRF, so they would be more comparable to our patients who at least have one CVRF, being that the disease itself. This is based on the well-reported knowledge that AGEs are related to inflammation and cardiovascular risk on the one hand and, on the other, that patients with autoimmune diseases such as rheumatoid arthritis, have an increased risk of cardiovascular disease that makes necessary to add a fixed multiplier of 1.5 to 2 to the established cardiovascular disease prediction general algorithms in order to adjust for the increased risk due to the disease [20]. Nienhuis et al. [14] selected a second control population with essential hypertension (EH), apart from the one conformed by HC. They found statistically significant differences in AGEs levels between SLE patients and HC but not between the SLE and the EH cohort, suggesting that finding differences when selecting HC with at least one CVRF could traduce a higher statistical power and a reduced probability of committing a type I error. Furthermore, they selected an SLE population with inactive disease, which might not reflect the reality of SLE patients in terms of disease characteristics in the way our patients might, which were included independently of their disease activity.

Additionally, we carefully examined all possible confounding factors to avoid drawing premature conclusions. Two controversial points were raised during the analysis. First, we observed only a positive trend shown by creatinine in the bivariate analysis of AGEs levels in the whole sample [21]. We discussed if that trend could have a fictitious origin since patients with SLE had higher creatinine levels (although in normal range) and were mostly located in the third AGEs tertile, and also since the trend was not observed when we analyzed the two groups separately. However, we finally decided to include creatinine in the model since there is ample evidence of a higher accumulation of AGEs in patients with renal failure [22] and lupus nephritis [12], and a difference could exist between groups since renal disease was an exclusion criterion in the HC group. Secondly, we found a negative association between dyslipidemia and AGEs, which was observed both in the combined analysis of the whole sample and in the HC separately (suggesting that such association comes from the HC group). The only data in the literature that could explain this negative association comes from the reported effect of lipid-lowering drugs in reducing AGEs levels [23]. Among HC, only 27 of the 85 with dyslipidemia (32%) were being treated with lipid-lowering agents, so we hypothesized that the rest could be controlling it with a lower-fat diet, which has also been associated with reduced AGEs levels [24]. Hence, we ended up including dyslipidemia in the model.

As for the interaction term between the main effect and dyslipidemia, although it was not found to be significant in the model, graphically the interaction seemed clear, especially in the group of SLE patients (Appendix A). This could be due to a lack of statistical power, since in the group of SLE patients there were only 8 dyslipidemic cases, unlike the 85 dyslipidemic HC. Therefore, the statistical power to detect this difference was much lower in the patient group, generating a less precise CI to reject the alternative hypothesis and leading to a lack of significance.

Regarding the study of AGEs relationship with SLE characteristics, we have found associations between AGEs levels and some disease activity indexes: SLEDAI, PGA, PtGA, CRP, and IL-6. As reflected in Section 2, the rise of AGEs levels with the increase of SLEDAI, which is the activity index most frequently used for SLE in clinical practice nowadays, showed a robust correlation. This association was also observed with other markers of activity commonly used to assess the disease state: PGA, PtGA, and IL-6. PGA is a part of the main indexes used currently to define remission or low disease activity in SLE. PtGA may be a more subjective parameter which can be influenced by external factors but that is clearly related to quality of life in SLE patients. IL-6 is not used routinely in the follow-up of SLE patients but its role in inflammation it is widely known generally and in rheumatic diseases in particular.

In the case of CRP, a significant association was only found between the upper tertile (0.28–3.92 mg/dL) and the first (<0.12), suggesting that the highest levels of AGEs were found among the patients with higher CRP values, both with values considered normal and abnormal (normal reference values in our laboratory <0.5 mg/dL). However, this correlation is only supported up to CRP values < 0.7 (R2 = 0.42, *p* < 0.0001), as graphically reflected in Appendix A. No correlation was found with higher CRP levels, which could be justified by a small number of patients with abnormal CRP levels. There was also a positive association with higher C4 levels, which draws attention since low C4 levels are the ones traditionally associated with high disease activity. However, although a decrease in complement levels is included in SLE classificatory criteria, there is wide controversy in the literature about the limited usefulness of the current techniques and types of complement measured in SLE and their ability to reflect disease activity [25]. Other uncertainties about complement are whether low levels should be persistent or combined (both C3 and C4) to be significant [26,27]. In our study, C3 levels showed a statistically significant direct correlation with C4 values (*p* ≤ 0.001) but not with AGEs levels. There was no association between having normal C4 levels at the moment of the study and not having had hypocomplementemia ever: 43% of the patients with current normal C4 levels had history of hypocomplementemia and 57% did not, while 77% of the patients with history of low C4 had now normal levels. This could traduce either fluctuant titers or normalized levels of C4 in response to treatment/lower disease activity and a need for further studies to elucidate the relation between complement and AGEs.

We also found a relationship between AGEs and indexes of accrual damage, the SDI. There is only a previous work in the literature that analyzed this association [15]. They found a correlation between AGEs and SDI in the univariate analysis that was lost after adjusting for age as well as in the multivariate analysis. In our case, the association persisted after adjusting for age and smoking status and any other possible confounding factor in the multivariate analysis. Considering this association, measuring AGEs levels could have a high impact in the prognosis of the disease helping to identify a subtype of patients with a more serious disease marked by higher accrual damage, which would be susceptible of a stricter follow-up and intensive treatment regimen, and subsequently allowing to improve these patients’ outcomes.

Specific manifestations (oral ulcers) or autoantibodies profile (less frequent anti-Ro60+ antibodies), could indicate a different clinical phenotype in SLE patients with less inflammation and thus, with lower AGEs levels. In clinical practice, it is very common to find overlaps of autoimmune diseases in the same patient, being especially frequent in SLE its overlap with Sjögren syndrome (SjS). It is known that both diseases have different inflammatory profiles [28], which could explain why there could be differences in AGEs levels between patients anti-Ro60 positive and negative. AGEs concentrations have been scarcely studied in SjS and efforts have not been directed to skin AGEs but RAGE and sRAGE with conflicting results [29,30,31], so more studies are needed to investigate AGEs levels in SjS and their differences both with SLE patients and with patients with a SLE-SjS overlap. Unfortunately, we could not validate this hypothesis in our study as the presence of SjS was recorded together with other autoimmune diseases as presence of overlapping syndrome in general, making studying the association only in SjS not possible. Furthermore, some patients had ongoing diagnostic SjS tests at the moment of our work. Similarly, oral ulcers are much more frequent in SLE than other autoimmune disease, potentially traducing a more typical SLE disease than in those without, which might justify differences in AGEs levels.

Regarding the negative relation found between AGEs and ANA antibodies, all patients were ANA+ at SLE diagnosis but 10 of them (8.2%) converted during disease follow-up and were ANA− at the moment of the study. It has been reported that the reduction of ANA responses might reflect the natural history of the disease as well as the effects of therapy [32]. Accordingly, these patients could have increased AGEs levels due to longer disease duration or more intense need for therapy due to more severe disease, and consequent more accrual damage and potentially higher AGEs levels. In our cohort, currently ANA− patients showed higher disease duration (15 vs. 10 years) and higher SDI (same levels of p25 and p50 but differences in p75: 1.56 vs. 0.68) although the differences were not statistically significant, probably due to lack of statistical power on account of the small sample size, also shown by the wide CI of this variable Appendix A. We didn’t observe differences in terms of taking immunosuppressants in the moment of the study between ANA+ and ANA− patients, but we did not retrieve data of the therapy history of patients, so we cannot rule out differences in the number of immunosuppressants or time taking therapy between both groups.

Despite the known relationship between AGEs and atherosclerosis, we did not find any correlation between AGEs levels and either CVRF or cardiovascular events (CVE). However, the *p*-value in the bivariate analysis was <0.1 and, considering that we have a small number of patients with CVE (N = 9), it is likely that our results are limited by a lack of statistical power which prevents us from drawing conclusions about the role of AGEs in cardiovascular risk. Furthermore, we assessed cardiovascular disease only through traditional CVRF or CVE and did not perform additional tests such as the intima-media thickness of the common carotid artery measured by ultrasound [15] or the small artery elasticity measured by pulse-wave analysis using tonometric recordings of the radial artery [14], both of which have been associated with AGEs levels in previous works. We also reassessed the correlation between AGEs and SDI excluding all variables related to cardiovascular disease (expressed as CVE in our study) as De Leeuw et al. do in their work [15]. They found a correlation in the bivariate analysis between skin AGEs and SDI, also after correction for the damage caused by CV disease. This association was not seen after adjusting for age or in the multivariate analysis. In our cohort, this new analysis did not alter the statistical correlation between SDI and AGEs, indicating that the association is not attributable to AGEs being associated to CV damage.

Only one of the two previous works studying skin AGEs in SLE have analyzed their association with disease characteristics, finding an association with age, creatinine, disease duration, the intima-media thickness of the common carotid artery, and the SDI in the univariate analysis, and only with age and disease duration in the multivariate one [15]. Our work has carried out a much more extensive analysis considering a great amount of demographic and clinical variables and performing a more complex statistical analysis considering all possible confounding factors, which provides a much deeper knowledge into these relationships and opens the door to the feasibility of using AGEs as a clinical tool for SLE management and prognosis.

Our study presents several limitations. Firstly, due to the retrospective nature of the study some data could not be retrieved such as the cumulative glucocorticoid (GC) dose that the patients had taken throughout the disease, and we could only assess the impact of GC through the current dose at the moment of the study. Likewise, the design makes it impossible to assess causality, which warrants future prospective studies. Secondly, and in order to clarify the effect on longstanding disease and therapy in AGEs levels, studies in newly diagnosed patients should be performed. Another limitation is that we did not check for all of the factors that have been described to influence AGEs levels such as diet [24].

To our knowledge, this is the second work to study and the first to find an association between SLE activity parameters and skin AGEs. We have found a correlation with, not one, but several SLE activity biomarkers and, also, with damage indexes. Furthermore, we have described, for the first time, skin AGEs associations with specific serological and clinical parameters that could define more precisely a specific type of patients in whom AGEs could have a particularly meaningful contribution. Therefore, our results are innovative and indicative of the promising role of AGEs and the AGEs skin reader as a tool to be implemented in daily clinical practice as a noninvasive, fast, real-time surrogate biomarker of SLE disease activity, damage, and specific manifestations.

## 4. Methodology

### 4.1. Subjects

This was a cross-sectional study conducted at the Hospital del Mar where patients of all ages who were visited at the SLE outpatient clinic, met the 1997 American College of Rheumatology (ACR) [33] or the 2012 Systemic Lupus International Collaborating Clinics (SLICC) classificatory criteria [34] for SLE, accepted to participate and signed the informed consent were randomly included. The exclusion criteria were pregnancy, diabetes mellitus (DM), treatment with corticosteroids at a dose equivalent to prednisone >20 mg/day, active malignancy, and fibromyalgia. Patients and the public were not involved in the design, conduct, reporting, or dissemination of this work.

### 4.2. Healthy Controls

The control population was selected from the ILERVAS cohorts (Vascular and Renal Translational Research Group, IRBLleida), which includes HC selected from primary care health centers, with at least one traditional CVRF and aged between 50 and 70 years if women or between 45 and 65 years if men. The traditional CVRF included were arterial hypertension (AHT) and/or dyslipidemia (DLP) and/or obesity (defined as a body mass index (BMI) > 30 kg/m^2^), and/or history in first-degree relatives of premature cardiovascular disease (men before 65-year-old and women before 60 years-old) and/or smokers and former smokers (<10 years since quitting). Exclusion criteria were as follows: history of cardiovascular disease (angina, myocardial infarction, cerebrovascular accident, peripheral arterial disease, intestinal ischemia or ischemia of some other territory), history of carotid surgery or surgery of arteries from other territories, DM and/or chronic renal disease (CRD), institutionalized population, population on long-term home-care, active neoplastic processes, life expectancy < 18 months [35]. AGEs levels were measured by autofluorescence in all of the HC.

### 4.3. Assessment of AGES Accumulation

In all patients, accumulated AGEs were measured non-invasively in the skin by an autofluorescence reader (Age Reader Mu Connect^®^, DiagnOptics Technologies BV, Groningen, The Netherlands) as described previously in the literature [10]. A light source emitting light at a wavelength of 320 to 400 nm excites fluorescent moieties in compounds in the skin to produce fluorescence at a wavelength of 420 to 600 nm (peak 440 nm). The output represents the ratio between autofluorescence in the range 420 to 600 nm and excitation light in the range 320 to 400 nm and is reported in arbitrary units (AU). Three consecutive AGEs measurements were taken from the ventral (anterior) surface of the forearm of each participant 10 cm below the elbow fold, avoiding any tattoos or heavily pigmented areas of skin. Measurements were performed at room temperature, while patients were in a seated position [36] (see Appendix A). The mean value of the three measures was calculated and compared with AGEs values from age-matched HC obtained from previous works [10].

### 4.4. Statistical Methods

#### 4.4.1. Comparison of Accumulated AGEs between Patients and Controls

A random sample of 60 individuals with systemic lupus erythematosus and of 183 healthy controls was calculated to be sufficient to estimate, with 95% confidence, a beta risk of 0.2 in a two-sided test, and an accuracy of ±0.25 units, the population mean of values (with an expected standard deviation of about 0.6 units [15]). HC were sex- and age-matched with a factor of approximately 3:1 to each of the SLE patients and selected according to the common variables between both groups. Due to the limited age range of our control group, some of the SLE patients had to be excluded as it was not possible to age-match them with HC. In addition, SLE patients with cardiovascular disease could not be included in the analysis due to it being an exclusion criterion in the HC sample. Difference of AGEs between SLE cases and HC was assessed through a fixed-effects analysis of covariance (ANCOVA) model adjusted for the confounding factors.

In order to identify potentially confounding variables, in addition to a bibliographic review about previously reported factors related to AGEs, a bivariate analysis was performed separating by cases and HC, and by tertiles of AGEs. Categorical data were described with absolute and relative frequencies, whereas continuous variables were displayed as mean (standard deviation), or as median (interquartile range) if non-normally distributed. In the case of categorical variables, we employed the Fisher’s exact test for variables with small frequencies and the χ^2^ test for the rest. For normal continuous variables, the Student’s *t*-test was used when analyzing two groups and the analysis of variance (ANOVA) when there were more than two. For non-normal continuous variables, the test used was the Mann-Whitney U test to compare two groups and the Kruskal-Wallis’ test to compare more than two. The significance level for these explorative analyses of confounding variables was taken to be <0.1.

Variables with statistically significant differences both between groups and with the AGEs response variable were considered potential confounders and were examined through interaction graphs before including them in the final model.

In the specific case of comparing AGEs levels between cases and controls and, as all of the HC were Caucasian, we performed a sensitivity analysis to assess the influence of ethnicity, testing only Caucasian patients against HC. We did not find any differences, so we kept all of the ethnicities in the final analysis. 

Later on, we explored the associations between AGEs levels (stratified in tertiles) and data of all of the participants of the study (both SLE patients and HC), in order to evaluate possible confounding factors. The bivariate analysis showed a significant positive relationship between smoking and AGEs levels, while creatinine showed a trend in that same direction. On the contrary, the presence of dyslipidemia was associated with lower values of AGEs (Appendix A).

According to these results and the differences found between SLE patients and HC, interaction graphs were created to visually assess smoking, age, dyslipidemia, and creatinine as cofounding variables. We found differences in the slopes of age and dyslipidemia (Appendix A) which were then evaluated in the fixed-effects analysis of covariance model (Appendix A). Smoking was also added to the model due to extensive literature linking it to AGEs values. Furthermore, in the smoking interaction graph we observed that the slopes of non-smokers and former smokers behaved similarly, with only a slight increase in mean cumulative AGEs in non-smokers with SLE, but apparently insignificant, so we unified non-smokers and former smokers in the same group vs. active smokers to increase statistical power (Appendix A).

According to all of the data explored, the multivariate model was adjusted with age, smoking, dyslipidemia, creatinine, and the interaction terms. None of the interaction terms were statistically significant so they were finally removed from the model except for the interaction between dyslipidemia and group (SLE or HC). This one, was not omitted since it allowed us to observe the effect (*p* = 0.062) of dyslipidemia, granting a better estimation of the AGEs value (Table 2). This was verified by adjusting it without the interaction, where the main effect of dyslipidemia was lost. Dyslipidemia was also adjusted for age and smoking (since HC with dyslipidemia were younger and smoked less), and its effect remained unchanged, ruling out that it was confused by other variables (Appendix A).

#### 4.4.2. Relation between Characteristics of SLE and Accumulated AGEs

An exploratory analysis was conducted using ANOVA tests adjusted for both age and current smoking status to investigate the association between SLE patient characteristics and the level of accumulated AGEs, including all patients from the cross-sectional study. For a better analysis, skewed variables of interest were categorized into tertiles or according to non-linear patterns, evaluated with general additive models. Associations with a *p* value < 0.1 were considered significant and, if consistent, were examined individually. First of all, the identification of potentially confounding variables was performed as described in the previous analysis (D.1.). Then multiple lineal regression models studying association between AGEs levels and each variable of interest were fitted considering the corresponding confounding factors, to avoid spurious associations. In this case, the significance level was taken to be <0.05.

In both analysis, continuous variables included in the final models were mean centered to facilitate interpretation. The assumptions of linearity, homoscedasticity and normality of the residuals were verified and the presence of influential points in each model was evaluated. All statistical work was carried out4 using R version 4.1.2.

## 5. Conclusions

SLE patients present higher skin AGEs levels than HC, supporting the hypothesis of the association between AGEs and SLE. Furthermore, the correlation observed between skin AGEs levels and SLE activity and damage markers indicate that AGEs seem to have a role as a new biomarker in this disease related to management and prognosis, which would have enormous implications in a field currently uncovered in SLE. The association with specific antibodies and disease manifestations may indicate a particular clinical phenotype related to higher AGEs levels, unveiling another potential clinical use of these products.

## Figures and Tables

**Figure 1 ijms-25-03022-f001:**
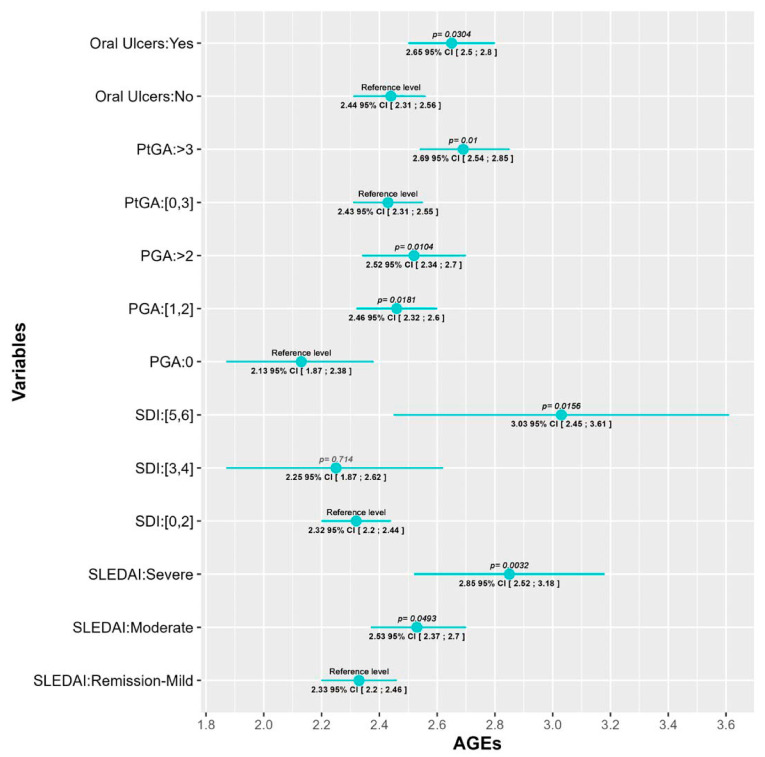
Statistically significant associations between AGEs levels and SLE characteristics and indexes. *p*-values < 0.05 (bold) indicate significant differences between the categories and the reference level of each variable; *p*-values not in bold indicate associations not statistically significant. PtGA: patient global assessment; PGA: physician global assessment; SDI: SLE damage index; SLEDAI: SLE disease activity index.

**Figure 2 ijms-25-03022-f002:**
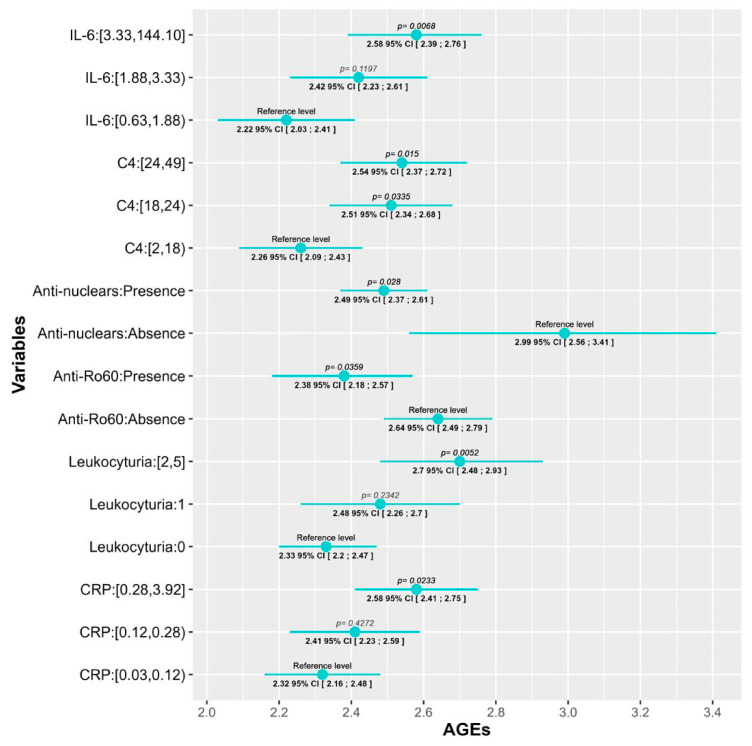
Statistically significant correlations between AGEs levels and SLE serological parameters. The change in AGEs values is depicted according to the reference category of each variable. *p*-value is considered significant if <0.05 (bold). IL-6: interleukin 6 (pg/mL); C4: complement 4 (mg/dL); CRP: C-reactive protein (mg/dL).

**Table 1 ijms-25-03022-t001:** Descriptive characteristics of cases and healthy controls and bivariate analysis between both groups. As we are exploring confounding variables *p*-value was widened and considered statistically significant if <0.1 (highlighted in bold in the text). AGEs: advanced glycation end products; HDL: High-density lipoprotein; LDL: Low-density lipoprotein.

	Controls	Cases	*p*-Value
	N = 189	N = 62	
Ethnicity			**<0.001**
Caucasian	189 (100%)	46 (74.2%)	
Other	0 (0.00%)	16 (25.8%)	
Age	56.0 [52.0; 62.0]	55.0 [51.0; 61.8]	0.193
Sex: Female	180 (95.2%)	58 (93.5%)	0.748
Hypertension	73 (38.6%)	14 (22.6%)	**0.032**
Obesity	61 (32.3%)	12 (19.4%)	**0.075**
Dyslipidemia	85 (45.0%)	9 (14.5%)	**<0.001**
Smoking			**0.054**
Never	79 (41.8%)	24 (38.7%)	
Former (>1 year)	54 (28.6%)	27 (43.5%)	
Active	56 (29.6%)	11 (17.7%)	
Body mass index	28.9 (5.98)	25.6 (4.65)	**<0.001**
Creatinine	0.70 [0.61; 0.77]	0.74 [0.64; 0.90]	**0.006**
Uric acid	4.90 (1.27)	4.70 (1.62)	0.365
Cholesterol	210 (37.5)	187 (39.5)	**<0.001**
HDL	61.9 (14.0)	65.9 (15.7)	0.125
LDL	138 (29.3)	112 (34.6)	**<0.001**
Triglycerides	123 [95.8; 160]	92.0 [70.0; 159]	**0.003**
Antidyslipidemics	27 (14.3%)	11 (17.7%)	0.649
Antihypertensives	61 (32.3%)	16 (25.8%)	0.424
AGEs	1.98 (0.45)	2.71 (0.56)	**<0.001**
AGEs in tertiles			**<0.001**
[1.0, 1.9)	83 (43.9%)	3 (4.84%)	
[1.9, 2.4)	74 (39.2%)	13 (21.0%)	
[2.4, 4.2]	32 (16.9%)	46 (74.2%)	

**Table 2 ijms-25-03022-t002:** Fixed-effects analysis of covariance (ANCOVA) model to study differences in AGEs levels between cases and healthy controls. y: years.

	Est.	2.5%	97.5%	t Val.	*p*-Value
Intercept	1.9418	1.8450	2.0385	39.5252	<0.0001
Group: Cases	0.7210	0.5660	0.8759	9.1645	<0.0001
Age (57.5 years)	0.0168	0.0081	0.0254	3.8359	0.0002
Smoking (Yes)	0.3265	0.1945	0.4585	4.8724	<0.0001
Creatinine (0.72 mg/dL)	0.2110	−0.1763	0.5983	1.0732	0.2843
Dyslipidemia (Yes)	−0.1240	−0.2544	0.0065	−1.8720	0.0624
(Group: Cases) + (Dyslipidemia (Yes))	0.1286	−0.2227	0.4799	0.7211	0.4715

**Table 3 ijms-25-03022-t003:** Variables that showed statistically significant differences according to AGEs tertiles in the bivariate analysis. M1: adjusted by age, M2: adjusted by age and smoking. “c” indicates variables which have been categorized as stated in Section 4. Bold indicates *p*-value < 0.1 and * indicates values according to the blood test performed in the study. *p*-val: *p*-value; SLEDAI: SLE disease activity index; SDI: systemic Lupus International Collaborating Clinics/American College of Rheumatology (SLICC/ACR) Damage Index; PGA: Physician global assessment; FACIT: Functional Assessment of Chronic Illness Therapy—Fatigue Scale; PtGA: Patient global assessment; GPT: Glutamic-pyruvic transaminase; CRP: C-reactive protein; IL-6: interleukin-6; ANA: antinuclear antibodies; C4: complement C4; GC: glucocorticoids; IS: Immunosuppressants (includes treatment with methotrexate, leflunomide, tacrolimus, mycophenolic acid or mycophenolate mofetil, azathioprine, cyclophosphamide, cyclosporine, rituximab or belimumab).

Variables	All	1st Tertile [1.2, 2.3)	2nd Tertile [2.3, 2.8)	3rd Tertile [2.8, 4.6]	*p*-Val M1	*p*-Val M2
	N = 122	N = 44	N = 41	N = 37		
Age	50.4 (14.9)	41.8 (13.8)	49.9 (12.2)	61.2 (11.9)		**<0.001**
Smoker	32 (26.2%)	10 (22.7%)	11 (26.8%)	11 (29.7%)	**<0.001**	
cDisease duration (years)					**0.082**	**0.090**
0–5	50 (41.0%)	19 (43.2%)	18 (43.9%)	13 (35.1%)		
6–10	16 (13.1%)	7 (15.9%)	6 (14.6%)	3 (8.11%)		
11–20	33 (27.0%)	13 (29.5%)	11 (26.8%)	9 (24.3%)		
>20	23 (18.9%)	5 (11.4%)	6 (14.6%)	12 (32.4%)		
Classificatory Criteria and Other Clinical and Serological Data						
Oral ulcers ever	50 (41.0%)	13 (29.5%)	18 (43.9%)	19 (51.4%)	**0.022**	**0.033**
Arthritis ever	92 (75.4%)	31 (70.5%)	32 (78.0%)	29 (78.4%)	**0.070**	**0.092**
Renal disease ever	8 (6.56%)	2 (4.55%)	1 (2.44%)	5 (13.5%)	**0.067**	**0.054**
cNumber of manifestations					**0.032**	**0.069**
[3, 7)	58 (47.5%)	19 (43.2%)	21 (51.2%)	18 (48.6%)		
7	24 (19.7%)	10 (22.7%)	8 (19.5%)	6 (16.2%)		
[8, 12]	40 (32.8%)	15 (34.1%)	12 (29.3%)	13 (35.1%)		
Disease Activity Indexes						
SLEDAI	4.00 [2.00; 6.00]	4.00 [0.00; 6.00]	4.00 [2.00; 6.00]	6.00 [2.00; 8.00]	**0.016**	**0.041**
cSLEDAI					**0.003**	**0.008**
Remission/Mild	71 (58.7%)	29 (67.4%)	25 (61.0%)	17 (45.9%)		
Moderate	39 (32.2%)	11 (25.6%)	14 (34.1%)	14 (37.8%)		
Severe	11 (9.09%)	3 (6.98%)	2 (4.88%)	6 (16.2%)		
SDI	0.00 [0.00; 1.00]	0.00 [0.00; 1.00]	0.00 [0.00; 1.00]	1.00 [0.00; 2.00]	**0.026**	**0.007**
cSDI_3					**0.052**	**0.017**
0–2	110 (90.9%)	41 (95.3%)	38 (92.7%)	31 (83.8%)		
3–4	8 (6.61%)	2 (4.65%)	2 (4.88%)	4 (10.8%)		
5–6	3 (2.48%)	0 (0.00%)	1 (2.44%)	2 (5.41%)		
PGA	2.00 [1.00; 3.00]	1.50 [1.00; 2.00]	2.00 [1.00; 3.00]	2.00 [1.00; 2.00]	**0.083**	**0.051**
cPGA					**0.051**	**0.029**
<1	18 (14.9%)	7 (16.3%)	6 (14.6%)	5 (13.5%)		
1–2	69 (57.0%)	27 (62.8%)	19 (46.3%)	23 (62.2%)		
>2	34 (28.1%)	9 (20.9%)	16 (39.0%)	9 (24.3%)		
Patient Reported Outcomes						
FACIT	17.5 [10.0; 27.0]	14.0 [9.00; 23.0]	22.0 [13.0; 30.0]	18.0 [10.0; 28.0]	**0.099**	**0.138**
PtGA	2.75 [1.00; 5.00]	2.00 [1.00; 3.00]	3.00 [2.00; 5.00]	3.00 [1.00; 5.00]	**0.028**	**0.042**
cPtGA					**0.112**	**0.121**
[0.0, 2.5)	57 (46.7%)	26 (59.1%)	14 (34.1%)	17 (45.9%)		
[2.5, 4.5)	28 (23.0%)	9 (20.5%)	12 (29.3%)	7 (18.9%)		
[4.5, 8.0]	37 (30.3%)	9 (20.5%)	15 (36.6%)	13 (35.1%)		
Serological variables						
GPT *	17.0 [13.0; 22.0]	16.0 [12.0; 22.5]	16.0 [13.0; 20.0]	18.0 [15.0; 23.0]	**0.095**	**0.068**
Total cholesterol *	181 (37.7)	172 (29.6)	174 (38.0)	201 (39.5)	**0.046**	**0.093**
cCRP *					**0.058**	**0.053**
[0.03, 0.12)	45 (37.2%)	24 (55.8%)	8 (19.5%)	13 (35.1%)		
[0.12, 0.28)	36 (29.8%)	11 (25.6%)	17 (41.5%)	8 (21.6%)		
[0.28, 3.92]	40 (33.1%)	8 (18.6%)	16 (39.0%)	16 (43.2%)		
cIL-6 *					**0.049**	**0.025**
[0.63, 1.88)	36 (33.3%)	18 (48.6%)	12 (31.6%)	6 (18.2%)		
[1.88, 3.33)	36 (33.3%)	11 (29.7%)	14 (36.8%)	11 (33.3%)		
[3.33, 144.10]	36 (33.3%)	8 (21.6%)	12 (31.6%)	16 (48.5%)		
ANA+ *	112 (92.6%)	43 (100%)	38 (92.7%)	31 (83.8%)	**0.027**	**0.036**
Anti-Ro60+ *	45 (37.8%)	17 (40.5%)	19 (47.5%)	9 (24.3%)	**0.183**	**0.164**
C4 *	19.8 (8.23)	18.5 (7.97)	18.7 (7.09)	22.4 (9.23)	**0.025**	**0.017**
Leukocyturia *	0.00 [0.00; 1.00]	0.00 [0.00; 0.00]	0.00 [0.00; 1.00]	1.00 [0.00; 2.00]	**0.004**	**0.001**
Hematuria *	0.00 [0.00; 0.00]	0.00 [0.00; 0.00]	0.00 [0.00; 0.00]	0.00 [0.00; 1.00]	**0.031**	**0.067**
cLeukocyturia *					**0.052**	**0.024**
0	72 (60.0%)	33 (78.6%)	24 (58.5%)	15 (40.5%)		
1	25 (20.8%)	6 (14.3%)	11 (26.8%)	8 (21.6%)		
[2, 5]	23 (19.2%)	3 (7.14%)	6 (14.6%)	14 (37.8%)		
Treatments						
GC	30 (24.6%)	7 (15.9%)	11 (26.8%)	12 (32.4%)	**0.004**	**<0.001**
Current dose of GC	5.00 [2.50; 10.0]	7.50 [3.75; 10.0]	5.00 [2.50; 12.5]	5.00 [2.50; 6.25]	**0.050**	**0.029**
Tacrolimus	1 (0.82%)	0 (0.00%)	0 (0.00%)	1 (2.70%)	0.147	**0.083**
cTreatment2					**0.077**	**0.092**
No IS	66 (54.1%)	27 (61.4%)	20 (48.8%)	19 (51.4%)		
IS	56 (45.9%)	17 (38.6%)	21 (51.2%)	18 (48.6%)		

## Data Availability

The data presented in this study are available on request from the corresponding author. The data are not publicly available due to ongoing research analysis.

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
