# Peer review of "Role of Advanced Glycation End Products as New Biomarkers in Systemic Lupus Erythematosus"

_ijms, 2024, doi:10.3390/ijms25053022_

Round 1
Reviewer 1 Report
Comments and Suggestions for Authors
I have reviewed the article titled "Role of advanced glycation end products as new biomarkers in systemic lupus erythematosus." A very interesting study in which the authors determine advanced glycation end products (AGE) at the skin level using autofluorescence between cases and controls and their correlation with clinical parameters.
The authors report that AGE levels in patients with SLE were significantly higher than in HC and that they can be used as biomarkers of the disease. In this regard, there are certain points that need to be addressed:
1. The introduction is very long, the authors should provide the most important background information that precedes their research.
2. In the introduction, the authors mention RAGE as a receptor for AGEs, but in the fourth paragraph, the authors put sRAGE, without mentioning it previously, the word sRAGE is referred to 7 times. It is not within the list of abbreviations nor is it described in the text. Does sRAGE refer to the same as RAGE? They are different?
The authors should put the meaning of this word or make evident the differences between sRAGE and RAGE
3. The authors should be more explicit about measuring AGEs in the skin and not in the serum. Were AGE levels compared with those obtained in serum from patients and controls? The authors must give a detailed explanation of why the determination in skin is better than in serum, since this is not included in the article.
4. The researchers indicate that the control subjects had a high body mass index and a high incidence of dyslipidemia, obesity and hypertension. It is worth mentioning that both obesity and dyslipidemia generate high concentrations of AGEs, how do the authors compare the AGEs of the controls with the AGES of the cases, if obesity by itself generates inflammation and high concentrations of AGEs?
5. Throughout the manuscript, the researchers refer to the statistical tests and types of statistical analysis, which they used to evaluate the results. It is worth mentioning that although statistical tests and analyzes are important since they focus on the presence or absence of significance, it is not necessary that these tests be mentioned in the results and discussion section, since at these points the results by themselves . themselves are the relevant ones and not the statistical tests used.
6. What is the statistical power of the study? The authors indicate that there is a "lack of statistical power" due to a low sample size.
7. Point 2.3 Variables is unimportant, since all research has variables, and these are introduced into the manuscript without referring to them in a special section, I suggest to the authors that this section be eliminated. Furthermore, in this section the authors speak as if it were a results section, weighing reference values and techniques. I suggest to the authors that this section be removed.
Comments on the Quality of English Language
The manuscript needs minor English revision.
Reviewer 2 Report
Comments and Suggestions for Authors
Carrion-Barbera I et al report on the potential use of AGEs as biomarkers in SLE.
Introduction is too long and would benefit from a reduction to the topics at hand - AGEs and SLE. The impression is of an thesis introductory chapter that has been copied and pasted with few modifications.
The authors appear to rely on only one method for the detection of AGEs.
In a submission with no less than 15 authors, it is particularly unhelpful for reviewers' and editors' time to be taken up with a manuscript which is missing all of the Tables and Supplementary Figures/Tables.
Generally speaking, the data (while highly processed) is of interest and a re-submission would be well received with the supplementary figure data.
The manuscript would benefit from careful review to remove errors e.g. in the Abstract at line 47- 51 there appears to be a sentence that has been split into two sentences.
Reviewer 3 Report
Comments and Suggestions for Authors
Dear authors, your main finding is the correlation between skin autofluorescence (functioning as a surrogate marker of skin reactive oxygen species) and biomarkers/clinical characteristics indicating an ominous SLE prognosis. Although the idea of using a non-invasive method to predict and detect SLE activity is intriguing, I have the following questions/observations:
Skin autofluorescence is influenced by multiple cofounders. Dyslipidemia, Smoking status (current, past - packyears), Coffee consumption, renal function, gender, and age are some of them. One has to normalize the dataset for the aforementioned cofounders. Is it possible for you to modify your analysis accordingly?
A more standardized approach could include a validation study comparing skin biopsies' ROS content (either indirectly through RAGE expression or directly through analytical methods). This would substantially strengthen your data and could prevent possible bias.
All of my best regards.
Round 2
Reviewer 1 Report
Comments and Suggestions for Authors
I have reviewed the corrected version of the manuscript "Role of advanced glycation end products as new biomarkers in systemic lupus erythematosus."
The authors have answered and corrected each of the suggested points. The article has become clearer, so I have no problem with it being published.
Author Response
We thank the reviewers for all their comments which have helped to increase the quality of our work.
Reviewer 2 Report
Comments and Suggestions for Authors
The manuscript has been significantly improved by incorporating reviewers' comments.
The supplementary figures/tables give confidence in the data.
The second sentence in the Discussion "This difference...stronger points" is not worded correctly. After the last comma it should probably read "our study having several stronger points". It may be more polite to say instead " and our research builds upon these studies in the following ways" but this is entirely the discretion of the authors.
Comments on the Quality of English Language
Satisfactory.
Author Response
We have change the sentence following the suggestion of the revierwer.
We thank the reviewers for all their comments which have helped to increase the quality of our work.
Reviewer 3 Report
Comments and Suggestions for Authors
Dear authors, this version of your work is substantially improved. I do not have further comments.
All of my best regards.
Author Response

(The authors gave the same response as above.)
